# The Antecedents of the Technology Acceptance Model in Microentrepreneurs' Intention to Use Social Networking Sites

**Rubinia Celeste Bonfanti** [1], **Francesco Tommasi** [2], **Andrea Ceschi** [2], **Riccardo Sartori** [2] **and Stefano Ruggieri** [1,*]

1 Faculty of Human and Social Sciences, Kore University of Enna, 94100 Enna, Italy; rubiniaceleste.bonfanti@unikore.it

2 Department of Human Sciences, University of Verona, 37129 Verona, Italy; francesco.tommasi@univr.it (F.T.); andrea.ceschi@univr.it (A.C.); riccardo.sartori@univr.it (R.S.)

* Correspondence: stefano.ruggieri@unikore.it

**Abstract:** Social media platforms offer significant growth opportunities for enterprises, particularly microenterprises, due to the chance to establish direct contact with customers. Drawing on the Technology-Acceptance Model (TAM), in the present study, we investigate the psychological reasons that lead microentrepreneurs to use Social Networking Sites (SNSs) for their business. In doing so, we also extend TAM by taking into account entrepreneurs' personalities (e.g., extraversion and openness to experience) and their perceived risk. We collected data by examining 247 microentrepreneurs engaged in the production of handmade objects. Our results confirm that of all the TAM behavioral antecedents tested, perceived usefulness and attitude toward SNSs' usage for business proved to be the best predictors of the intention to use SNSs for business activity. The results also indicate that extraversion, openness to experience, and perceived risk, as external factors, significantly affect the TAM constructs. We discuss implications and suggestions for future research.

**Keywords:** entrepreneurship; social networking sites; technology-acceptance model; perceived usefulness; attitude; perceived risk; personality

## 1. Introduction

Due to the widespread use of Information and Communication Technology (ICT) to organize our lives and society, entrepreneurs have extensively embraced new technologies in recent years [1]. ICT has now become a vital element of entrepreneurial strategies since it can offer opportunities for firms to achieve their objectives and generate innovative ideas. Notably, ICT is typically linked to various aspects of a business plan, e.g., marketing and management strategies, the kinds of goods and services provided, and the company's technological infrastructure [2–5].

In this context, the literature has focused on specific ICT, namely Social Networking Sites (SNSs), which are virtual platforms that can provide benefits to entrepreneurs. For example, SNSs enable improved internal organization, enhanced economic performance, and improved customer engagement [6–8]. SNSs are especially helpful to microentrepreneurs (microenterprises are defined as enterprises that employ fewer than 10 persons and whose annual balance sheet total does not exceed EUR 2 million [9]) because they enable them to make new connections [10,11], build a business relationship with their clients [12,13], enhance decision-making procedures, and enhance communication with their clients, leading to customer loyalty.

Even with these advantages, many microentrepreneurs choose in-store shops and other conventional sales methods to expand their number of consumers and their business. Such a preference could be due to the fact that entrepreneurial innovation opportunities can be influenced by various environmental, behavioral, and cognitive factors, which should

be examined individually [14]. Despite these initial thoughts, it is still unclear which psychological reasons could influence this choice.

The overarching aim of the present study is to describe the effects of potential psychological variables that may influence microentrepreneurs' behavioral intention to use SNSs (BIuSNS) in their business activity. To pursue our aim, we developed an empirical study following theoretical insights of an extended version of the Technology-Acceptance Model (TAM) [14]. Our objective was to identify the predictive variables for microentrepreneurs' BIuSNS for their businesses. In this way, the present study offers two main contributions to the literature. First, the study contributes by providing an empirical test of the viability of a model predicting SNSs use in entrepreneurs. Second, the study also supports the development of a theoretical framework for applying the TAM to the entrepreneurial context of social media.

## 1.1. Entrepreneurs' Psychological Aspects and Social Networking Site Usage

As noted, SNSs enable interaction and information sharing between users and groups. Recruiting customers and receiving feedback from them through comments and likes make SNSs extremely important for microentrepreneurs [15–18]. These tools enable entrepreneurs to develop their brand and business through interactions and contacts with investors, workers, consumers, and suppliers with a low-cost approach [19–21]. Despite these financial advantages, many microentrepreneurs prefer a business strategy based on in-store shops (i.e., stores situated inside the town, in suburban areas, or inside retail centers) [22,23].

The decision to employ in-store shops rather than using new technologies appears to be driven more often by individual preferences than by a smart business strategy. Indeed, a number of studies have shown that when an entrepreneur launches a business activity, s/he brings his/her human capital to the company, making it a reflection of their way of life [24–26]. All of these characteristics emphasize the value of psychological approaches in research on how entrepreneurs use SNSs for their business.

According to the literature, entrepreneurs' acceptance and perception of the usefulness of technology (see the TAM), personality characteristics (extraversion and openness), and perceived risk are crucial elements of how they employ technologies in doing business [27]. First, the TAM has emerged as the most successful among various theories examining entrepreneurs' behavior [28,29]. Originating from behavioral psychology, TAM was developed to model users' acceptance of ICT and to identify the causal relationships between users' internal attitudes, behavioral intentions, and beliefs about technology. The TAM suggests that the acceptance of new technology is connected with two particular beliefs: (a) perceived usefulness and (b) perceived ease of use. The former is a person's subjectively determined likelihood that utilizing a technology system will result in performance improvement. Perceived ease of use is the extent to which a user of a certain application system believes that using it would be effortless on both a physical and mental level [15]. Taken together, perceived usefulness and perceived ease of use affect an individual's attitude toward using a technological system [15]. The TAM has been used in a large number of studies to examine people's acceptance of a variety of technologies, including word processors, the World Wide Web, e-mail, e-learning environments, and online commerce [30–34].

Second, an entrepreneur's personality also appears to be an important aspect of entrepreneurial behavior and SNS usage. Because the Big Five personality theory has been shown to explain the variability in overall internet use more effectively, we have employed external variables belonging to this theory to extend TAM [35]. Particularly, we considered extraversion and openness to experience traits from the Big Five personality theory. The personality trait that has been found to have the greatest impact on the tendency for creativity is openness to new experiences [36,37]. High levels of openness to new experiences are characterized by curiosity, innovation, and freedom from convention [37,38]. Numerous studies have demonstrated the direct relationship between openness to experience and in-

novative behaviors, including general internet use [35], the desire to employ virtual reality teams [39], and individual innovativeness in information systems [39,40]. For example, Amichai–Hamburger and Vinitzky [41] showed that individuals who are open to new experiences are more likely to use various SNS features, probably because they are naturally intrigued and eager to try new things. Another essential quality for entrepreneurs, according to the Big Five personality theory, is extraversion. This trait determines the ease of developing social networks, acquiring outside resources, and establishing strong networks. Additionally, it has been observed that those with high degrees of extroversion are more likely to find online social services (such as talking and meeting new people) on the internet to be valuable and useful [42]. According to Zhao and Seibert [43], extraversion is a fundamental feature that affects the decision to use various social connection channels, such as SNSs [44,45], and it influences the propensity for sociability.

Last, perceived risk is a predictor of technology acceptance [46–49]. Perceived risk is defined as a subjective impression of an objective risk based on information, prior experiences, and/or intuitive assessment [50]. Understanding risk perception might be useful in predicting how entrepreneurs react to difficulty and novelty in the management of their business (this is still a major emphasis of current perceived-risk research) [51,52]. The risk dimension within the TAM appears as a negative precursor for choices involving technology, namely in the general adoption of technology [53–55], in adopting mobile banking [56], in predicting online shopping intention [57], and for other choices involving technology [58,59]. Specifically, it has often been shown as a predictor of perceived usefulness [48,60], but it is also a direct determinant of behavioral intention [61]. However, it is still unknown how and to what extent risk perception has a significant impact on entrepreneurial behavior and SNS usage. Given its unquestionable influence on enterprises' strategic and operational decisions, it is important to consider risk in investigating entrepreneurs' decision-making dynamics [62].

### 1.2. The Study

Following our argument, the present study aims to provide empirical evidence on the viability of a TAM model integrated with the dimensions of perceived risk, openness to experience, and extraversion as external factors driving microentrepreneurs' BIuSNS. Understanding the role of psychological variables as predictors of microentrepreneurs' BIuSNS can help us to understand the entrepreneurial behavior relating to the use of SNSs and enable the adoption of more expansive interpretative models. Figure 1 elaborates on the extended model on which we based our hypotheses.

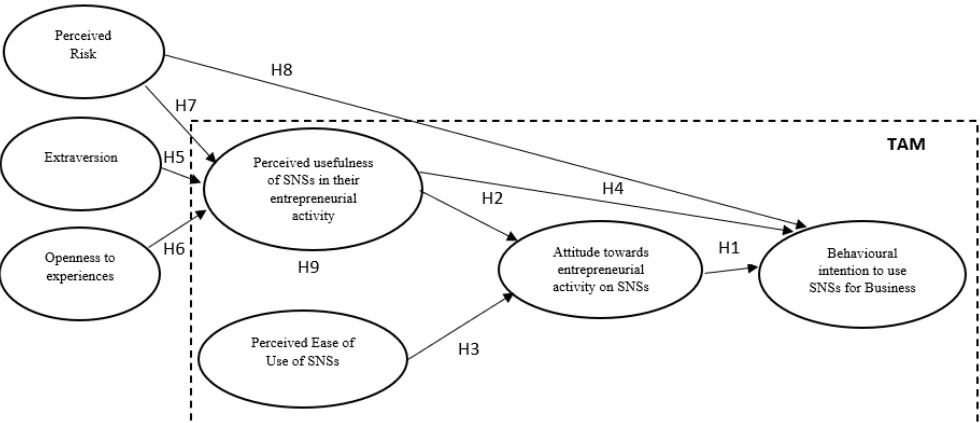

**Figure 1.** Graphical depiction of the extended model of the TAM for predicting behavioral intention to use SNSs.

Narrowly, we expect an effect of the variables belonging to the TAM in the context of microentrepreneurs' BIuSNS for their business. To test the psychological pro-

cesses underlying the entrepreneurs' choices, based on the TAM [15,63], we propose the following hypotheses:

**Hypothesis 1**: *Attitude toward entrepreneurial activity on SNSs positively influences a microentrepreneur's BIuSNS for business.*

**Hypothesis 1a**: *Attitude toward entrepreneurial activity on SNSs has a mediating role between perceived usefulness and BIuSNS for business.*

**Hypothesis 1b**: *Attitude toward entrepreneurial activity on SNSs has a mediating role between perceived ease of use and BIuSNS for business.*

According to the TAM [15,63], behavioral intention to use SNSs for one's business is a result of attitude toward entrepreneurial activity on SNSs, which is in turn influenced by the perceived usefulness of SNSs (PU-SNS) and the perceived ease of use of SNSs (PEU-SNS). Finally, PU-SNS is a direct influencer of BIuSNS for business. Therefore, we expect that:

**Hypothesis 2**: *PU-SNS for entrepreneurial activity significantly influences attitude toward entrepreneurial activity on SNSs.*

**Hypothesis 3**: *PEU-SNS significantly influences attitude toward entrepreneurial activity on SNSs.*

**Hypothesis 4**: *PU-SNS for entrepreneurial activity positively affects BIuSNS for business.*

Coupling H1–4 with current research on the relationship among personality factors, perceived risk, and entrepreneurs' perception of technologies, we further hypothesize that:

**Hypothesis 5**: *Extraversion has a positive relationship with PU-SNS.*

**Hypothesis 6**: *Openness to experience has a positive relationship with PU-SNS.*

**Hypothesis 7**: *Perceived risk has a negative relationship with PU-SNS.*

**Hypothesis 8**: *Perceived risk negatively affects BIuSNS for business.*

Lastly, the literature on the TAM informs that perceived usefulness is an intervening dimension that acts between external factors and individuals' intentions [64]. Özbek et al. [42] reported the mediating role of perceived usefulness in the relationship between personality and behavioral intention to use smartphones. Similarly, Im et al. [48] reported that perceived usefulness plays a mediating function in the relationship between perceived risk and the behavioral intention to adopt technologies. Based on these pieces of evidence, we proposed the following hypothesis:

**Hypothesis 9**: *PU-SNS has a mediating role between attitude and the following external variables: perceived risk (H9a), extraversion (H9b), and openness to experience (H9c).*

## 2. Materials and Methods

### 2.1. Participants and Procedure

A total of 247 microentrepreneurs (men = 138, women = 109) between 22 and 68 years old (M = 40.36, SD = 10.51) who do not use SNSs for business participated in the study. We randomly selected participants from the local Chamber of Commerce listings in three medium-sized southern Italian cities who were directly informed about the research by an active association of microentrepreneurs in southern Italy. All the business owners involved were engaged in the production of handmade objects. They created and traded

jewelry, clothing, accessories for the home and body, presents, souvenirs, etc. We asked 300 participants to complete an anonymous online survey, and the response rate was 82.3%.

### 2.2. Measures

**Sociodemographic Data**. For sociodemographic data, we asked participants to report their age and gender (1 = male, 2 = female, 3 = other).

**Attitude toward Entrepreneurial Activity on SNSs**. We assessed participants' attitudes toward entrepreneurial activity on SNSs with the semantic differential measurement technique [65]. Utilizing a 7-item scale (e.g., bad/good, ugly/beautiful, weak/strong), respondents assessed the target phrase, "For you, a business activity carried out exclusively through social networking sites is . . . ". Each item was displayed on a 5-point scale ($\alpha = 0.92$), with the left side representing a negative term and the right side representing a positive one.

**PU-SNS in Entrepreneurial Activity**. We measured PU-SNS by adapting three items from Davis [15] (e.g., "Using social networking sites is useful in my job"). Each item was displayed on a 5-point Likert scale (1 = strongly disagree, 5 = strongly agree; $\alpha = 0.85$).

**PEU-SNS in Entrepreneurial Activity**. We measured PEU-SNS by adapting six items from Davis [15] (e.g., "Learning to use social networking sites would be easy for me"). Each item was displayed on a 5-point Likert scale (1 = strongly disagree, 5 = strongly agree; $\alpha = 0.88$).

**Extraversion and Openness to Experience**. We measured microentrepreneurs' extraversion and openness to experience using the Big Five questionnaire [66,67]. The participants answered 48 questions on a 5-point Likert scale (1 = strongly disagree, 5 = strongly agree; $\alpha_{extraversion} = 0.83$, $\alpha_{openness} = 0.84$).

**BIuSNS for Business**. We measured microentrepreneurs' BIuSNS for business with a single item. The statement was, "I intend to start using SNSs for my business activity". The item was presented on a 5-point Likert scale (1 = strongly disagree, 5 = strongly agree).

**Perceived Risk to Use SNSs for Business**. We measured microentrepreneurs' perceived likelihood of using SNSs for business with three commonly used perceived risk categories in marketing literature [68]: financial, performance, and psychological. Items included, "It is probable that using social networking sites for my business activity would not be worth its cost". The items were presented on a 5-point Likert scale (1 = strongly disagree, 5 = strongly agree; $\alpha = 0.78$).

We refined the instruments through a pilot session after ten university students filled out the questionnaire during statistics lessons, looking for any unclear or inaccurate questions.

We assessed the survey's reliability using Cronbach's alpha and the average variance extracted (AVE) test [69]. Results from Cronbach's alpha are comparable to those from other studies, and the values range from 0.78 to 0.92, demonstrating a high level of reliability (exceeding the minimum recommended level of 0.6). The AVE results also show a high degree of reliability (exceeding the minimum recommended level of 0.5) [70].

### 2.3. Data Analytic Plan

We tested our hypotheses using structural equation modeling (SEM). Figure 1 shows the theoretical model tested in this study. We conducted model testing using Mplus software, v. 7.0. To measure the SEM and test our hypotheses, we used the bootstrapping technique with 5000 resamples [71] and assessed the overall goodness-of-model fit using the $\chi 2$ statistics ($\chi 2/df$ ratios < 3 indicate reasonable fitting models), the comparative fit index (CFI, with values > 0.90 indicating better fitting models) [72], and the root-mean-square error of approximation (RMSEA; values < 0.08 indicate good fit) [72].

## 3. Results

Table 1 reports descriptive statistics of the study variables. We conducted preliminary analyses to verify the normality of the data distribution [73]. All the variables of the study indicated no significant deviation from normality in the data distribution (|Skewness| < 1).

**Table 1.** Means, standard deviation, skewness, and kurtosis of the variables.

| Variables | Mean | SD | Skewness | Kurtosis |
|---|---|---|---|---|
| Age | 40.36 | 10.51 | 0.962 | 0.797 |
| Attitude toward SNSs | 3.88 | 1.99 | 0.563 | −0.135 |
| Perceived usefulness of SNSs | 3.92 | 1.68 | 0.453 | −1.376 |
| Perceived ease of use | 3.33 | 1.52 | 0.353 | 0.376 |
| Extraversion | 37.47 | 4.90 | 0.954 | 0.122 |
| Openness to experiences | 39.42 | 3.23 | 0.834 | 0.143 |
| Perceived risk | 3.66 | 1.63 | 0.372 | 0.436 |
| Behavioral intention to use SNSs | 3.97 | 1.58 | 0.440 | −1.233 |

Note: SNSs = Social Network Sites.

*Mediation Analysis*

Our SEM model showed a good fit ($\chi 2$ = 42.724, df = 24, $\chi 2$/df = 1.8, CFI = 0.979, RMSEA = 0.051, 95% RMSEA = 0.026–0.079), accounting for 21% ($R^2$ = 0.196) of the variance in behavioral intention.

Overall, our data supported all the hypotheses regarding the extended TAM model except H3. As Table 2 shows, the relationship between attitude and BIuSNS is significant ($\beta$ = 0.387, $p < 0.05$), supporting H1. This is the same for H2, with the association between PU-SNS and attitude being significant ($\beta$ = 0.371, $p < 0.05$). Unexpectedly, the relationship between PEU-SNS and attitude is not significant ($\beta$ = 0.067, $p$ = 0.217), which led us to reject H3. Conversely, the relationship between PU-SNS and BIuSNS is significant ($\beta$ = 0.438, $p < 0.05$); as well as the relationship between extraversion and PU-SNS is significant ($\beta$ = 0.401, $p < 0.05$), and the relationship between openness and experience and PU-SNS is significant ($\beta$ = 0.391, $p < 0.05$), supporting H4–6. Finally, the relationships of perceived risk with PU-SNS and BIuSNS are significant ($\beta$ = −0.452, $p < 0.05$; $\beta$ = −0.399, $p < 0.05$, H7–8).

**Table 2.** Structural model and effect size.

| Variables | β-Value | SE | *p*-Value |
|---|---|---|---|
| *Direct paths* | | | |
| Attitude → behavioral intention | 0.387 | 0.018 | 0.000 |
| Perceived usefulness → attitude | 0.371 | 0.021 | 0.000 |
| Perceived ease of use → attitude | 0.067 | 0.045 | 0.217 |
| Perceived usefulness → behavioral intention | 0.438 | 0.047 | 0.035 |
| Extraversion → perceived usefulness | 0.401 | 0.096 | 0.000 |
| Openness to experience → perceived usefulness | 0.391 | 0.085 | 0.000 |
| Perceived risk → perceived usefulness | −0.452 | 0.075 | 0.000 |
| Perceived risk → behavioral intention | −0.399 | 0.093 | 0.002 |
| *Indirect paths* | | | |
| Extraversion → perceived usefulness → attitude | 0.139 | 0.007 | 0.002 |
| Openness to experience → perceived usefulness → attitude | 0.128 | 0.011 | 0.012 |
| Perceived risk → perceived usefulness → attitude | −0.117 | 0.021 | 0.007 |
| Perceived usefulness → attitude → behavioral intention | 0.318 | 0.507 | 0.000 |
| Perceived ease of use → attitude → behavioral intention | 0.024 | 0.041 | 0.327 |

In addition, Table 2 reports the indirect/mediating role of PU-SNS and attitude in the assumed external variable (extraversion and openness to experience). In this regard, the relationship among extraversion, PU-SNS, and attitude ($\beta$ = 0.139, $p < 0.05$) and the relationship among openness to experience, PU-SNS, and attitude ($\beta$ = 0.128, $p < 0.05$) are positive and significant; finally, the relationship among perceived risk, PU-SNS, and

attitude (β = −0.117, *p* < 0.05) are negative and significant. Therefore, H9a, H9b, and H9c are supported.

The relationship among PU-SNS, attitude, and BIuSNS (β = 0.318, *p* < 0.05) is positive and significant. This supports H1a, and PU-SNS has a mediating role between extraversion and attitude. Conversely, the association among PEU-SNS, attitude, and BIuSNS (β = 0.024, *p* < 0.327) is not significant, leading to the rejection of H1b.

Our results partially showed a significant association with all of the TAM model's variables. Specifically, attitude and PU-SNS proved to be the best predictors of the microentrepreneurs' BIuSNS business activity of all the TAM behavioral antecedents tested.

## 4. Discussion

In this study, we aimed to shed light on the relationship between entrepreneurship and SNS use for businesses, highlighting the role of the psychological antecedents of this phenomenon in the theoretical framework of the TAM. To better understand the relationships between BIuSNS for business, attitude toward entrepreneurial activity on SNSs, PEU-SNS, PU-SNS in entrepreneurial activity, and three external variables (extraversion, openness to experiences, and perceived risk), we tested a mediation model. Regarding the TAM indicators, our results demonstrated a substantial correlation with all of the model's variables only partially. Specifically, attitude and PU-SNS in entrepreneurial activity influenced microentrepreneurs' BIuSNS business as expected.

Our results show that PU-SNS has a direct relationship with entrepreneurs' BIuSNS for business and an indirect relationship through attitudes. Surprisingly, the indirect relationship among PEU-SNS, attitude, and BIuSNS for business is not significant, indicating that PEU-SNS does not play a fundamental role in microentrepreneurs' BIuSNS. This finding contradicts previous investigations [74] and could be explained because after the pandemic, using technological devices has become easier for all people [75–78]; therefore, the decision to use them for business activity might be prescinded from PEU-SNS.

Regarding the direct and indirect relationship between PU-SNS and microentrepreneurs' BIuSNS for business, our findings imply that PU-SNS plays a crucial role in the understanding of this relationship, exerting a direct influence on the BIuSNS and an attitude-mediated effect. Therefore, this variable plays a crucial role in the intention to use or continue to use SNSs for business. The perceived usefulness is likely related to opportunity capability or capacity to identify market opportunities using a variety of methods [79–81]. Opportunity competency is one of the essential traits of entrepreneurs and one of the more crucial and distinctive skills for knowledgeable entrepreneurs [82]; it stands for the capacity to look for, identify, develop, and assess any opportunity in a specific market [83]. Entrepreneurs may only succeed in their businesses and minimize risks by identifying opportunities [84,85]. Specifically, those who perceived the usefulness of SNSs changed their previous attitudes toward the use of SNSs in their business activities. This indirect path shows greater awareness and probably stems from the entrepreneur's conviction regarding SNSs' usefulness for their business. Therefore, this is a process that goes beyond mere economic utility and leads to a behavioral intention to use SNSs for business. Additionally, even when attitudes remained the same, the PU-SNS continued to have a significant impact on microentrepreneurs' BIuSNS for business, probably because, in this case, the BIuSNS is driven by purely economic motivation.

In our extended TAM, we hypothesized that openness to experience would have a relationship with PU-SNS, and our study's results confirmed this hypothesis. This implies that individuals' characteristics, such as curiosity, imagination, willingness to entertain new ideas, and open-mindedness [67], are strictly linked to the PU-SNS for their own business. Several studies have shown that openness turns out to be closely related to behaviors involving innovation, such as general internet use [35], the intention to use virtual reality teams [86], and personal innovativeness in information systems [39]. Regarding the specific case of social media, one study showed that people who are more innovative, curious,

and creative than others are more likely to find utility in all SNSs [40,87]. Our study also confirmed these results.

Moreover, we hypothesized that extraversion would have a relationship with PU-SNS, and our results confirmed this hypothesis. Studies have suggested that extroverted entrepreneurs normally show dominant traits in making decisions that also influence their propensity for innovation [88]. It has been seen that individuals with high levels of extroversion have a higher propensity to believe that online social services are useful [42]. In this regard, several studies have shown that extraversion is closely related to behaviors involving innovation, such as the use of either information or leisure services on the internet [89], online information sharing [90], and smartphone usage [64]. Regarding the specific case of social media, one study showed that people who are more optimistic and enthusiastic than others use Facebook [87] and find all the SNSs attractive [44]. Indeed, SNSs would be another way for extrovert entrepreneurs to be more assertive and, therefore, could be seen as very helpful for business owners, pushing them toward the BIuSNS.

Finally, we hypothesized that perceived risk would have a relationship with PU-SNS and BIuSNS for business. Our study's results confirmed this hypothesis. The study's results suggest a negative effect of perceived risk on PU-SNS for entrepreneurial activity. Moreover, participants with lower scores on PU-SNS are less inclined to use SNSs for their business. Studies have also shown that perceived risk is a key element of human decision-making processes [91,92] because people evaluate all of an activity's advantages (positive outcomes) and hazards (negative outcomes) before deciding whether to do it. For example, according to Pavlou [58], risk is seen as an indirect precursor of the behavioral intention of e-commerce practices, and it has a negative impact on the behavior because of the online environment's virtual nature and implicit uncertainty. Furthermore, Nam and Lee [59] investigated variables affecting consumers' adoption of Internet banking. According to their findings, risks have a negative impact on one's desire to use new technology, whereas perceived usefulness and attitude have a positive impact. Other authors achieved similar outcomes when they discussed the potential risks and acceptability of new technologies [58,93]. Our results confirm that perceived risk is a negative predictor of BIuSNS for business. In conclusion, the acceptance of new technologies is expected to be constrained by rising risk perception.

Ultimately, the present study enriches the application of the TAM by combining it with the Big Five personality theory and a perceived risk construct to the field of entrepreneurship and identifying individuals' psychological antecedents in the cognitive evaluation process of the BIuSNS for business. These empirical results also allow for the identification of a psychological profile of microentrepreneurs who are more inclined to utilize SNSs for business purposes, which is helpful for market planning in the business innovation sector.

*Limitations and Future Research*

Despite our theoretical implications, our study presents certain limitations. First, the participants self-reported the psychosocial variables, and they might have been influenced by response biases, such as social desirability. To mitigate potential biases, it is recommended that researchers make every attempt to use behavioral measures. Second, although we obtained a randomized sample from a specific group, we only included a few people from a particular geographic region, which might not be a representative sample of the entire community. A comparison of various territorial realities, particularly from a perspective of economic development, may offer further information about the relationships examined in this study. Another limitation is given by common method bias, which implies that a portion of the variance in our study may have been due to the methods used. Future research needs to identify more valid instruments and procedural strategies to minimize common method bias. Finally, we limited our research to microentrepreneurs in the handmade product sector. In further analysis, researchers should investigate the approach suggested here in small and medium business owners, as well as in different categories of products.

## 5. Conclusions

Overall, the study clarified the multifaceted and underexamined relationship between microentrepreneurs and SNSs. What stands out in particular is the crucial influence that the TAM factors (PEU-SNS and PU-SNS), personality, and perceived risk in explaining this relationship. Based on this study, important roles are given to attitude, PU-SNS, openness to experience, extraversion, and perceived risk. Contrary to expectation, the BIuSNS for one's business is unaffected by PEU-SNS.

These results also have practical implications. E-commerce conducted on SNSs is assuming a crucial role in bridging infrastructure gaps and granting opportunities for everyone, especially microentrepreneurs, to access global markets effortlessly. In this sense, we have shown that perceiving new technology as a risk is the first obstacle entrepreneurs must overcome. Moreover, perceiving SNSs' usefulness as a tool for improving one's business is the first of the helpful components for changing personal behavior, which increases the desire to use SNSs for business.

**Author Contributions:** Conceptualization, R.C.B. and S.R.; methodology, S.R.; formal analysis, R.C.B.; writing—original draft preparation, F.T. and R.C.B.; writing—review and editing, A.C., F.T., R.S. and S.R. All authors have read and agreed to the published version of the manuscript.

**Funding:** This research received no external funding.

**Institutional Review Board Statement:** This study was conducted in accordance with the Declaration of Helsinki and approved by the Ethics Committee of Psychological Research of the Faculty of Human and Social Sciences, University of Enna (approval code: UKE-IRBPSY-07.20.1).

**Informed Consent Statement:** Informed consent was obtained from all subjects involved in the study.

**Data Availability Statement:** The data presented in this study are available upon request from the corresponding author.

**Acknowledgments:** We thank the microentrepreneurs who agreed to participate in this study.

**Conflicts of Interest:** The authors declare no conflict of interest.

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
