# Peer review of "The Antecedents of the Technology Acceptance Model in Microentrepreneurs’ Intention to Use Social Networking Sites"

_ejihpe, doi:10.3390/ejihpe13070096_

Round 1

Reviewer 1 Report

Dear authors:

the review:

The Antecedents of the Technology Acceptance Model in  Entrepreneurs’ Intention to Use Social Networking Sites

In the  study authors  investigate the psychological reasons that lead microentrepreneurs to use SNSs for their business. 

PLUS:   very good structured article

Introduction - Methods - DIscussion - (Tabs- graphs) - Conclusions

results: s confirm that of all the TAM behavioral antecedents tested, perceived usefulness and attitude toward SNSs’ usage for business proved to be the best predictors of the intention to use SNSs for business activity.

line 365-366: "Based on this study, a particular role is
assigned to attitude, perceived usefulness, openness to experience, extraversion, and perceived risk. On the contrary, perceived ease of use does not influence the decision to use SNSs for one’s business."

line. 212 add to discussion

- important is studying (receive skills) at university

Tkacová, H.; Králik, R.; Tvrdoň, M.; Jenisová, Z.; Martin, J.G. Credibility and Involvement of Social Media in Education—Recommendations for Mitigating the Negative Effects of the Pandemic among High School Students. Int. J. Environ. Res. Public Health 2022, 19, 2767. https://doi.org/10.3390/ijerph19052767

Ben-Shalom, U., Hajaj, C., Davidovitch, N., & Berger, C. (2023). Online Temporary Learning Groups in Higher Education – Interactions, Compensation, and Maximisation of Achievements in an Israeli Case Study. Journal of Education Culture and Society, 14(1), 619–633. https://doi.org/10.15503/jecs2023.1.619.633

Sirotkin, A.V.; Pavlíková, M.; Hlad, Ľ.; Králik, R.; Zarnadze, I.; Zarnadze, S.; Petrikovičová, L. Impact of COVID-19 on University Activities: Comparison of Experiences from Slovakia and Georgia. Sustainability 2023, 15, 1897. https://doi.org/10.3390/su15031897

Tkácová, H.; Pavlíková, M.; Stranovská, E.; Králik, R. Individual (Non) Resilience of University Students to Digital Media Manipulation after COVID-19 (Case Study of Slovak Initiatives). Int. J. Environ. Res. Public Health 2023, 20, 1605. https://doi.org/10.3390/ijerph20021605

excellent: analyses and IDEAS (congratulation) : Variables : line: 262

I am persuaded  that your article will have many citations.

I recommand publishing your article.

very good job !

Author Response

Response to Reviewer 1 Comments

PLUS:   very good structured article

Response: Thank you!

Introduction - Methods - Discussion - (Tabs- graphs) – Conclusions

Point 1 - results: confirm that of all the TAM behavioral antecedents tested, perceived usefulness and attitude toward SNSs’ usage for business proved to be the best predictors of the intention to use SNSs for business activity.

Response: Thank you for the suggestion. We have added a confirm in result section, as follows (RL 271-274):

“Our results partially showed a significant association with all of the TAM model’s variables. Specifically, attitude and perceived usefulness proved to be the best predictors of the microentrepreneurs’ intention to use SNSs for business activity of all the TAM behavioral antecedents tested”

Point 2 - line 365-366: "Based on this study, a particular role is assigned to attitude, perceived usefulness, openness to experience, extraversion, and perceived risk. On the contrary, perceived ease of use does not influence the decision to use SNSs for one’s business."

Response: Thank you for the suggestion. We have revised the paragraph, as follows (RL 383-386):

“Based on this study, important roles are given to attitude, perceived usefulness, openness to experience, extraversion, and perceived risk. Contrary to expectation, the choice to use SNSs for one's business is unaffected by perceived ease of use.”

Point 3 - line. 212 add to discussion: - important is studying (receive skills) at university

Response: Thank you for the suggestion. We have revised the paragraph, as follows (RL 220-222):

“We refined the instruments through a pilot session after ten university students filled out the questionnaire during statistics lessons, looking for any unclear or inaccurate questions. “

Tkacová, H.; Králik, R.; Tvrdoň, M.; Jenisová, Z.; Martin, J.G. Credibility and Involvement of Social Media in Education—Recommendations for Mitigating the Negative Effects of the Pandemic among High School Students. Int. J. Environ. Res. Public Health 2022, 19, 2767. https://doi.org/10.3390/ijerph19052767

Ben-Shalom, U., Hajaj, C., Davidovitch, N., & Berger, C. (2023). Online Temporary Learning Groups in Higher Education – Interactions, Compensation, and Maximisation of Achievements in an Israeli Case Study. Journal of Education Culture and Society, 14(1), 619–633. https://doi.org/10.15503/jecs2023.1.619.633

Sirotkin, A.V.; Pavlíková, M.; Hlad, Ľ.; Králik, R.; Zarnadze, I.; Zarnadze, S.; Petrikovičová, L. Impact of COVID-19 on University Activities: Comparison of Experiences from Slovakia and Georgia. Sustainability 2023, 15, 1897. https://doi.org/10.3390/su15031897

Tkácová, H.; Pavlíková, M.; Stranovská, E.; Králik, R. Individual (Non) Resilience of University Students to Digital Media Manipulation after COVID-19 (Case Study of Slovak Initiatives). Int. J. Environ. Res. Public Health 2023, 20, 1605. https://doi.org/10.3390/ijerph20021605

Response: Thank you for the suggestion. We have added these interesting studies in our references.

excellent: analyses and IDEAS (congratulation)

Response: Thank you!

Reviewer 2 Report

Overall, it is a well-crafted study with reasonable theoretical foundations coming fundamentally from TAM studies [14] and the study on the influence of founder personalities, behaviors and new venture success in Sub-Saharan Africa [25]. Because of this, this study contributes to potential TAM extensions and to better understanding of micro-entrepreneurs’ intention to use social networking sites. While this study qualifies for publication and could be published as it is, I consider that it could be improved as follows:

1.     The title should be more accurate, given that this study results apply only to micro-entrepreneurs and not to entrepreneurs in general. Suggest replacing “entrepreneurs” with “microentrepreneurs” in the title.

2.     The definition of microentrepreneurs for the purpose of the study is not provided. It is expected that the local chamber of commerce used to provide the contacts must have a criterion to classify their members into microentrepreneurs and others. Important to provide the definition of your target of study.

3.     The response rate of 82.3% for anonymous online survey is extremely unusual in this type of studies. Notice that the study of the Sub-Saharan entrepreneurs got 18.4% while expected response rates in the U.S. rank between 5% and 15% (being the last a very successful response rate). Should you comment on this, it would be helpful.

4.     Extraversion, a key independent variable, is not properly defined and the rationale for its role in the model is not provided. Furthermore, the discussion (lines 312-316) are convincing respect to its impact on innovation while the fact that extraverted people are more optimistic and enthusiastic about the use of SNS is still not the same as thinking it will increase perceived usefulness (why not perceived easy of use? Why not moderating the paths of H2 and H3?). A better argumentation is needed for this hypothesis.

5.     A good discussion of why openness to new experiences is more likely to use various SNS features is provided; however, the rationale for expecting an effect on “perceived usefulness” (rather than directly on attitude towards entrepreneurial activity on SNSs) is not provided prior to the proposal of the hypothesis. It may be convenient to provide some argumentation in section 1.2 study rather than waiting till discussion (lines 302-310).

6.     Authors may consider Post Hoc analyses for future studies, in which the 3 psychological variables: perceived risk, extraversion and openness to experiences be considered moderators of the paths H2 and H3. While the proposed paths for the TAM model extension may make sense, the reader is left with the idea there are alternative hypotheses/explanation. In particular, due to the failure to obtain significant results in H3, one of the basic tenets of the TAM model.

7.     Most of these studies have a limitation given by common method bias. Authors should try some of the basic tests for this issue or at least acknowledge that this is a limitation, shared by most of survey studies due to the complexity of addressing it properly, of most studies of this type.

8.     Finally, one minor detail is that TAM didn’t originate from social psychology (which requires more than one person to participate in the social phenomenon). TAM was developed using the theory of reasoned action (for one individual) as its basis and this is in the area of behavioral psychology.

Overall, this study contributes with the addition of the psychological variables: perceived risk, extraversion and openness to extend the TAM model as well as using this extended model to understand the antecedents of the use of social network sites by microentrepreneurs. The above comments are intended to improve the quality of the study.

Author Response

Response to Reviewer 2 Comments

Overall, it is a well-crafted study with reasonable theoretical foundations coming fundamentally from TAM studies [14] and the study on the influence of founder personalities, behaviors and new venture success in Sub-Saharan Africa [25]. Because of this, this study contributes to potential TAM extensions and to better understanding of micro-entrepreneurs’ intention to use social networking sites. While this study qualifies for publication and could be published as it is, I consider that it could be improved as follows:

Point 1: The title should be more accurate, given that this study results apply only to micro-entrepreneurs and not to entrepreneurs in general. Suggest replacing “entrepreneurs” with “microentrepreneurs” in the title.

Response: Thank you for your comment. In the revised version of the manuscript, we have added a new title, as follows:

“The Antecedents of the Technology Acceptance Model in Microentrepreneurs’ Intention to Use Social Networking Sites”

Point 2: The definition of microentrepreneurs for the purpose of the study is not provided. It is expected that the local chamber of commerce used to provide the contacts must have a criterion to classify their members into microentrepreneurs and others. Important to provide the definition of your target of study.

Response: Thank you for your comment. In the revised version of the manuscript, we have added a definition of microentrepreneurs as follows (page 2):

“Microenterprises are defined as enterprises that employ fewer than 10 persons and whose annual balance sheet total does not exceed EUR 2 million”.

Point 3: The response rate of 82.3% for anonymous online survey is extremely unusual in this type of studies. Notice that the study of the Sub-Saharan entrepreneurs got 18.4% while expected response rates in the U.S. rank between 5% and 15% (being the last a very successful response rate). Should you comment on this, it would be helpful.

Response: Thank you for the comment. We have explained a possible reason of our high response rate as follows (RL 181-183):

“We randomly selected participants from the local Chamber of Commerce listings in three medium-sized southern Italian cities, who were directly informed about the research by an active association of microentrepreneurs of southern Italy.”

Point 4: Extraversion, a key independent variable, is not properly defined and the rationale for its role in the model is not provided. Furthermore, the discussion (lines 312-316) are convincing respect to its impact on innovation while the fact that extraverted people are more optimistic and enthusiastic about the use of SNS is still not the same as thinking it will increase perceived usefulness (why not perceived easy of use? Why not moderating the paths of H2 and H3?). A better argumentation is needed for this hypothesis.

Response: Thank you for the suggestion. We have added some references in the introduction and the discussion to underline the rationale for its role in the model, as follows (RL 104-106; 328-332):

“Additionally, it has been observed that those with high degrees of extroversion are more likely to find the online social services (such as talking and meeting new people) on the internet to be valuable and useful”.

“It has been seen as individuals who have high levels of extroversion had a higher propensity to believe useful the online social services on the internet. In this regard, several studies have shown that extraversion is closely related to behaviors involving innovation, such as the use of either information or leisure services in the Internet, online information sharing, and smartphone usage”.

Point 5: A good discussion of why openness to new experiences is more likely to use various SNS features is provided; however, the rationale for expecting an effect on “perceived usefulness” (rather than directly on attitude towards entrepreneurial activity on SNSs) is not provided prior to the proposal of the hypothesis. It may be convenient to provide some argumentation in section 1.2 study rather than waiting till discussion (lines 302-310).

Response: Thank you for the suggestion. We have added some studies in section 1.2 of the study to underline the rationale for its role in the model, as follows (RL 96-101):

“Numerous studies have demonstrated the direct relationship between openness to experience and innovative behaviors, including general Internet use, the desire to employ virtual reality teams, and individual innovativeness in information systems. For example, Amichai-Hamburger and Vinitzky showed that individuals who are open to new experiences are more likely to use various SNS features, probably because they are naturally intrigued and eager to try new things”.

Point 6: Authors may consider Post Hoc analyses for future studies, in which the 3 psychological variables: perceived risk, extraversion and openness to experiences be considered moderators of the paths H2 and H3. While the proposed paths for the TAM model extension may make sense, the reader is left with the idea there are alternative hypotheses/explanation. In particular, due to the failure to obtain significant results in H3, one of the basic tenets of the TAM model.

Response: Thank you for the suggestion. We have already begun a study in which we will consider the Post Hoc analyses that you have suggested.

Point 7: Most of these studies have a limitation given by common method bias. Authors should try some of the basic tests for this issue or at least acknowledge that this is a limitation, shared by most of survey studies due to the complexity of addressing it properly, of most studies of this type.

Response: Thank you for the suggestion. We have added this limitation as follows (RL 372-375):

“Another limitation is given by common method bias, which implies that a portion of the variance in our study may have been due to the methods used. Future researches need to identify more valid instruments and procedural strategies to minimize common method bias.”

Point 8: Finally, one minor detail is that TAM didn’t originate from social psychology (which requires more than one person to participate in the social phenomenon). TAM was developed using the theory of reasoned action (for one individual) as its basis and this is in the area of behavioral psychology.

Response: Thank you for the suggestion. We have modified our sentence as follows (RL 75-77):

“Originating from behavioral psychology, the TAM was developed to model users’ acceptance of ICT and to identify the causal relationships between users' internal attitudes, behavioral intentions, and beliefs about technology“.

Overall, this study contributes with the addition of the psychological variables: perceived risk, extraversion and openness to extend the TAM model as well as using this extended model to understand the antecedents of the use of social network sites by microentrepreneurs. The above comments are intended to improve the quality of the study.

Response: Thank you!
